# Effects of Shot Peening and Cavitation Peening on Properties of Surface Layer of Metallic Materials—A Short Review

**DOI:** 10.3390/ma15072476

**Published:** 2022-03-27

**Authors:** Aleksander Świetlicki, Mirosław Szala, Mariusz Walczak

**Affiliations:** Department of Materials Engineering, Faculty of Mechanical Engineering, Lublin University of Technology, 20-618 Lublin, Poland; m.walczak@pollub.pl

**Keywords:** cavitation peening, shot peening, peening, cavitation, surface treatment, metal alloy, hardness, roughness

## Abstract

Shot peening is a dynamically developing surface treatment used to improve the surface properties modified by tool, impact, microblasting, or shot action. This paper reviews the basic information regarding shot peening methods. The peening processes and effects of the shot peening and cavitation peening treatments on the surface layer properties of metallic components are analysed. Moreover, the effects of peening on the operational performance of metallic materials are summarized. Shot peening is generally applied to reduce the surface roughness, increase the hardness, and densify the surface layer microstructure, which leads to work hardening effects. In addition, the residual compressive stresses introduced into the material have a beneficial effect on the performance of the surface layer. Therefore, peening can be beneficial for metallic structures prone to fatigue, corrosion, and wear. Recently, cavitation peening has been increasingly developed. This review paper suggests that most research on cavitation peening omits the treatment of additively manufactured metallic materials. Furthermore, no published studies combine shot peening and cavitation peening in one hybrid process, which could synthesize the benefits of both peening processes. Moreover, there is a need to investigate the effects of peening, especially cavitation peening and hybrid peening, on the anti-wear and corrosion performance of additively manufactured metallic materials. Therefore, the literature gap leading to the scope of future work is also included.

## 1. Introduction

The performance of metallic materials such as wear resistance can be improved by applying the appropriate surface treatment. One of the traditional methods of increasing resistance to abrasive wear is so-called shot peening (SP). However, the development of shot peening results in introducing new treatment methods such as cavitation peening (CP). Therefore, it seems essential to characterise the up-to-date peening processes and review their effects on the properties of a surface layer of metal alloys. Currently, the additive manufacturing of metallic components is considered rather modern when it comes to manufacturing technology [1,2,3]. However, the surface of additively constituted parts needs to be finished, and in particular, shot peening can be easily employed. Therefore, this review paper paid particular attention to additively manufactured metallic materials.

One of the most promising peening methods is cavitation peening [4,5,6]. The literature systematically studies these peening methods. The crucial application of cavitation peening is the removal of stresses after previous treatments. Exemplary fabrication processes such as milling, forging, or surfacing introduce additional onto the surface layer of the material [7,8], which is harmful to the operational performance of machine components. Therefore, peening can reduce the rate of stress induced by deformation, known as strain hardening. This is the blocking (shortening) of the free path of a dislocation that is in motion in a given plane of the easy slip [9]. The continuous shortening of the free path causes an increase in the plastic flow resistance, which is due to an increase in dislocation density in the peened metals [10]. It is known that heat treatment modifies ductility, hardness, change phase composition, and the grain size of metallic components [11,12]. Therefore, shot peening is considered a competitive process to heat treatment, enabling the required mechanical properties of steel [13,14]. Although, it should be pointed out that, even though most metallic materials can be successfully processed by peening, the limited types of metal alloys undergo heat treatment.

There is a broad range of peening processes reported in the up-to-date literature, and cavitation peening seems to have the highest scientific and industrial potential, which is followed by the literature survey results given in Table 1. An analysis of search results was conducted in Scopus and Web of Science databases for the phrases ‘cavitation peening’, ‘shot peening’, and ‘cavitation peening + shot peening’. There is a noticeable increase in the number of publications in all categories however; according to the information contained in the graphs, the number of publications is the highest for SP, with about 4000 publications, while cavitation peening belongs to a narrower research area, and about 200 publications were devoted to it.

Recently, much attention has been paid to additive technologies. However, unfortunately, characteristic surface morphology and microstructure features of additively manufactured components can deteriorate the mechanical properties, which consequently need to be improved [15,16,17]. For this reason, various peening methods have been used for finishing machining parts made of metallic materials produced by printing, and combinations of peening methods with other methods, such as heat treatment, are also used [18]. Since parts manufactured using additive manufacturing techniques often have complex shapes, the innovation of additive manufacturing techniques is also important [19]. The surface post-treatment of complex shape components can be efficiently conducted using different shot peening methods. Significantly, the use of cavitation peening seems to make the most sense, due to the possibility of processing locations for working fluid where solid shots cannot reach. Moreover, it is suggested that the combination of shot peening methods, namely hybrid peening in the case of additive manufactured components and the use of lightweight metals such as magnesium alloys and aluminium could pave the way for new, unprecedented solutions by increasing the flexibility of the design process [20]. Unfortunately, limited papers on the peening of additive manufactured metallic materials and literature regarding the hybrid peening of metallic materials is also scant.

This paper reviews the peening processes and characterizes the effects of the shot peening and cavitation peening treatments on the surface layer properties of metallic components. First, the paper’s layout introduces the reader to peening methods. Then, it characterizes the effects of shot peening (SP) and cavitation peening (CP) on metallic materials. Next, the SP and CP applications for additive manufactured metallic materials are analysed. Finally, the literature gaps leading to the scope of future work are proposed.

## 2. Shot Peening

Shot peening is a method that has been used since the 1950s. The method was initially used mainly for aviation-related applications. However, it has spread to other industries [21]. Shot peening can improve properties such as the development of surface roughness and hardness [22], improving the resistance to corrosion [23], fatigue [24,25], rolling contact fatigue [24], and fretting [26].

The peening process is carried out in the following manner, and particles with high hardness are accelerated in compressed air or even stream to an appropriate speed, and move towards the component to be burnished. This results in collisions of the particles with the workpiece and other particles. Due to the high hardness of the incident particles, when they hit the surface, their kinetic energy is converted into the plastic deformation of the workpiece. Indentations are produced in the material, and a plastic deformation zone and an elastic zone are formed in the vicinity [21].

During shot peening, the rearrangement of dislocations takes place, which moves from the interior to the vicinity of the grain suit. As the local density of dislocations near the grain boundaries increases, their imbalance also increases. The basis of shot peening is to increase polycrystals’ free energy and generate more defects and interfaces (grain boundaries) through various nonequilibrium processes. Shot peening is a commonly used method to improve the surface layer properties of components, machinery, and equipment. However, shot peening is not a method without disadvantages, including high Ra values, which can partially be removed by grinding or polishing. There are also surfaces that shots cannot reach due to the complex shape of the component [27,28].

To understand the shot peening process, it is vital to know the factors that influence the process. Figure 1 shows the main parameters and factors affecting the process. These are divided into three main subgroups: those related to the medium, the peening equipment, and those related to the work piece. Often, the following are cited as the most important parameters: the peening intensity (frequency) and the area/surface of the shot peening coverage. Many components are subjected to the shot peening process, including turbine blade tips, connecting rods, gears, shafts, axles, springs, and torsion bars. Necessary materials suitable for SP can include high strength steels, aluminium alloys, or isothermally hardened ductile cast iron, and recently also additive manufactured components. Other applications for SP include flutter forming and contour correction. Figure 2 shows a comparison of different surface modification methods.

## 3. Peening Methods

There are many methods of coring as well as classifications of these methods. Therefore, this paper focuses on the treatments selected; valid, current, and innovative methods were selected. The shot peening methods described below are also important due to their frequent use, as well as their continuous development and possible further applications.

### 3.1. Ultrasonic Impact Peening

The method was developed in 1970 in the Soviet shipbuilding industry but was later commercialised under the name Esonix UIT [31]. This method uses an electro-mechanical transducer that generates ultrasonic waves, which are then processed by a piezoelectric or magnetostrictive transducer, and a resonator is set in motion [31,32]. Ultrasonic impact peening is also called ultrasonic hammer peening or direct sonotrode-driven hammer peening. The tool used in this method is a steel pin driven in high-frequency oscillations by an ultrasonic generator. Deburring takes place through the interaction of the tool with the workpiece. The high-frequency impacts create indentations in the material, and thus the material is plastically deformed [33]. Figure 3 shows a comparison of the typical depth of influence between shot peening, laser shock peening, deep rolling, and hammer peening.

### 3.2. Ultrasonic Nanocrystal Surface Modification

In 2000, a new pressing technique called ultrasonic nanocrystalline surface modification (UNSM) was patented by Pyoun due to the ease of creating a nanocrystalline structure [34]. UNSM is a method of material modification that involves peening the surface using movements of a tool that oscillates at a very high frequency. In a typical UNSM process, a tungsten carbide (WC) tip or Si_3_N_4_ ball is attached to an ultrasonic horn that strikes the surface up to 20,000 times per second at 1000 to 10,000 shots per square millimetres. To prevent the overheating of the shot peened workpiece, oil is used and sprayed on the workpiece [35,36].

The UNSM method has some advantages over other shot peening methods. The main advantage of this method is the possibility of extensive control of the process, i.e., applying a higher or lower static load, and changing the speed of oscillation by changing the amplitude of vibrations. Thus, this method allows high repeatability and the selection of ideal parameters for specific conditions. The method allows for grain fragmentation, introducing residual compressive stresses, reducing surface defects, and increasing hardness [37].

The UNSM method can be used to treat components after additive manufacturing, as demonstrated by Min Seob Kim et al. They investigated the behaviour of 316 L steel after direct energy deposition (DED) fabrication, after optimising the process, improving the roughness corrugation, and increasing the microhardness by 72%, 80%, and up to 71.2%, respectively [38]. Studies have also been conducted for titanium and its alloys [39], aluminium alloys after 3D printing [40], as well as 304 steel [41] or chromium-molybdenum steel [42].

### 3.3. Laser Shock Peening

The laser shock peening technology was developed at the Battelle Institute, dating back to the late 1960s. The laser shot peening (LSP) method uses pulses of laser light to generate shock waves that are expected to deform the material and leave residual compressive stress [28]. Laser light and a transparent layer are required, although a non-transparent layer is also often used. The temperature at the heating point reaches up to 10,000 °C and the pulse length is about 0.15 to 0.30 ns. LSP can be performed in two configurations, direct and limited ablation. A transparent layer (usually water) is used to reduce the plasma produced on the surface and increase the shock wave’s intensity. After the laser light beam passes through the transparent layer, it hits an opaque layer, usually made of aluminium, zinc, or copper, which is also called a sacrificial coating. The opaque layer is used to prevent direct laser ablation on the surface and melting [43]. An impulse of laser light reaches the absorption layer and then ionisation occurs in the heated zone, and a shock wave is generated, which plastically deforms the material [44]. The resulting compresive residual stresses (CRS) can reach up to 1.5 mm below the surface, depending on the method used [45]. Figure 4 shows different surface severe plastic deformation (SPD) treatments and the corresponding plastically deformed top surface of the target material.

### 3.4. Flap Peening

In this method, balls are placed on two or more flaps which are fixed on a shaft. The tool is then made to rotate, which transfers energy to the balls, and the tool is brought close to the surface so that the balls begin to strike the material, thus deforming it, leaving numerous indentations [47].

### 3.5. Micro Shot Peening

Micro shot peening is carried out using minimal spherical particles called fine-particle or micro shot. The main difference from the SP is the sizes of shots that range from 0.03 to 0.15 mm in diameter. Generally, the micro shots are made of high-speed tool steel, cemented carbide, glassy alloy and ceramic, with high hardness [48].

It has been found that micro shot peening improves the scuffing resistance of 17CrNiM steel, and a combination of SP and micro shot peening gives even better results [49]. Other studies indicate better roughness, hardness, and CRS parameters compared to SP, and the combined treatment of SP and micro shot peening also improved all of these values compared to SP [50]. An improvement in fatigue strength for railway axle steels of 25% compared to the unpenned surface has also been observed [51]. Research is also being conducted on titanium alloys regarding the influence of shot size on material properties and corrosion resistance after the peening process [52].

### 3.6. Water Jet Peening

In this method, the pressing medium is high-velocity water droplets hitting the surface. The droplets generate a high peak pressure which plastically deforms the surface. Water jet peening is a term used in many other peening methods, but it should be noted that it is different from cavitation peening. Therefore, in order to avoid misunderstandings, the classification used by H. Soyama was adopted. Water jet peening refers to methods using water jets, where water droplets are the pressing medium, and the mechanism is the collision of liquids. On the other hand, cavitation peening is a method using cavitation impacts, and the material deformation mechanism is due to shock waves. The problem of dividing these methods is probably due to the fact that, in classical cavitation peening, cavitation is induced by a jet of water injected at high speed into the water [53,54]. Results on spring steel, titanium, and nickel suggest that this may be a valuable method for applications where increased roughness and CRS are required [55].

### 3.7. Oil Jet Peening

The principle of this method is the same as for water jet peening, the difference being the medium, which is hydraulic oil, used to create the jet. With this method, it is possible to introduce CRS without significant changes in the surface topography. The maximum surface CRS value after oil jet peening can be 80–85% of the yield point, while for shot peening, it is 45–97%. For shot peening, the surface CRS is 30–80%, for water jet peening, it is 30–60%, and for cavitation water jet peening, it is about 89% of the yield strength [27,56,57]. By using oil jet peening, it is possible to improve CRS in materials such as low carbon steel or aluminium alloys.

### 3.8. Ultrasonic Shot Peening

This method uses powerful ultrasonic vibrations of a hard body with a high frequency, usually ultrasonic, which set spherical shots in motion. As a result, the intensity of pressing is very high. The main difference lies in the absence of the medium setting the shots in motion (in the form of air or steam). There are also shot collisions, so there is no uniform angle of incidence on the component as in the case of SP, where the angle is 90°, and there is no ball impact force [58]. Studies comparing ultrasonic shot peening (USP) and SP suggest two approaches that can be considered. Tekada found similar structural and mechanical properties for USP and SP, but concluded that under similar conditions, a larger volume of nanocrystallites was observed after SP. However, a different conclusion was reached by Dai and Shaw, who stated that, because the shot size used during USP can be larger than for SP, the unit impact is able to deliver more energy, despite the lower particle velocity [59]. According to Tekada, the coverage area in his condition was 170%/s and the particle velocity was >100 m/s for SP. For USP, the coverage area was 20%/s and the particle velocity < 20 m/s with a very large velocity distribution in contrast to USP. Despite many attempts, it has not been possible to provide a consistent model for multiple impacts [60]. Research in this area suggests that the type and composition of the peening balls, among other things, influence the intensity, but to a lesser extent than the length of peening time. USP gives good results for soft materials and reduces the probability of crack initiation and propagation. Positive aspects have also been found for the physical and mechanical properties of such treatment, such as improved corrosion resistance, increased contact, and fatigue strength [61].

## 4. Cavitation Peening

Cavitation peening does not use shots but produces cavitation, and the deformation is caused by pressure waves and microflows from collapsing bubbles and cavitation clouds [53]. As in the case of water jet peening, this method does not leave any contamination in the material in the form of shot particles, and the process can be controlled [62]. A great advantage of cavitation peening is the absence of a shot peening medium that could contaminate the surface, as in other methods, e.g., SP, where ceramic or metallic particles are used, or, as in the case of LSP, where surface contaminating products may appear. Differences between cavitation peening and water jet peening (WJP) also include the relationship between injection pressure and process capability. In the case of WJP in the range from 20 to 60 MPa, the process capability also increases with increasing pressure. The situation is different in the case of CP, where this relation has a parabolic shape in the same range, i.e., it increases to a value slightly below 40 MPa, and then decreases. The process efficiency at 40 MPa is three times higher for cavitation peening than for water jet peening [53]. This method is effective for peening hard materials such as titanium and its alloys [63,64]. Chromium-molybdenum steel is also a topic for research due to its common use in industrial applications [65,66,67].

### 4.1. Cavitation

The name cavitation—used in physics and technology—is derived from the Latin word ‘cavitas’ (emptiness, cavity). Cavitation is a rapid physical transition from a liquid to a gaseous state due to a decrease in pressure, which is associated with the appearance in the liquid of the so-called cavitation bubbles with huge implosion loads [68,69,70]. Bubbles filled with steam, dissolved gases or both steam and gases are formed in places with a pressure reduced to a critical value and implode in places with increased pressure. The time of collapse of the bubbles may be below 0.001 s.

The gas contained in the cavitation bubble causes the bubble to be reconstituted after implosion, and this process can repeat several times [71,72]. The determining factor for the formation of cavitation is the changing pressure field. There must be a region in the flow where the pressure drops to a value close to the saturated vapour pressure and then rises. The factors influencing the formation of cavitation are pressure, temperature and velocity of the fluid flow. Other factors that facilitate the cavitation phenomenon include the surface shape with which the liquid contacts the presence of impurities and gases dissolved in the liquid [73].

Osborn Raynolds first described the phenomenon of cavitation in 1894. It can be observed in liquids with a non-uniform velocity distribution. If the fluid rapidly increases its velocity, the static pressure of the fluid decreases. The boiling point of a fluid is closely related to the pressure surrounding the fluid. As the pressure decreases, the boiling point also decreases. If the fluid’s static pressure drops low enough, the fluid starts to boil and thus, cavitation bubbles form [74,75]. Figure 5 shows the collapse of a cavitation bubble, i.e., cyclic changes in bubble size with wave propagation in the liquid, followed by an implosion, i.e., a sudden reduction in bubble size leading to its disappearance. Implosion causes vapour to condense inside the bubble, and the implosion pressure ranges from several hundreds to several thousands of mega Pascals. Important for understanding the mechanism of material deformation during cavitation is the formation of microbubbles. If the collapsing bubble is in contact with the solid surface, the diameter of the microstructure can be approximately 10% of the bubble radius, and its velocity will be high enough to deform the solid surface [29]. If the asymmetric collapse of the bubble is at larger distances from the solid surface, there will be a significant reduction in its failure [76,77,78].

The formation of liquid microbubbles may also be due to the fact that the pulsating cavitation cloud interacts with spherically shaped bubbles close to the solid. These interactions cause the bubbles to oscillate, and their shape becomes unstable. If the amplitude of the bubble oscillation is large enough, microstructure formation begins to occur as the liquid surrounding the bubble begins to flow through the bubble towards the solid. The velocity of the micro-streams can range from 156 to 175 m/s, according to various sources [79,80]. Gibson has shown that jet formation, direction, and intensity are functions of the bubble–bubble interaction [81].

As a result of the interaction of bubbles in the cloud, the cloud collapse process experiences a slight deceleration in the primary phase and significant acceleration in the final phase of implosion. This results in the generation of a much higher final pulse pressure than is apparent from the analysis of the implosion of a single bubble [83,84]. The velocity of the micro stream produced by the interaction with pressure pulses resulting from the collapse of other bubbles and the propagation of the shock wave can be as high as 400 m/s [85].

Many types of cavitation can be considered, and successive authors have proposed their own divisions of this phenomenon. The most popular and, at the same time, the simplest division is the one proposed, among others, by Lauterborn [78]. Considering the causes of cavitation, we can distinguish the following types of this phenomenon [86]:optical cavitation;molecular cavitation;acoustic cavitation;hydrodynamic cavitation.

#### 4.1.1. Hydrodynamic Cavitation

Cavitation can be induced in many ways; however, hydrodynamic cavitation seems to be the most common type used for cavitation peening treatment [73]. Hydrodynamic cavitation can occur in the Venturi tube, orifice, or nozzle components, as a result of local pressure drop caused by various constrictors or by the mechanical rotation of the vortex diode and other rotating-type devices [87]. As an example, when high-pressure water is injected into the water, it is produced in the core of the vortex in the shear layer around the jet where the pressure drops; in addition, the vortex cavitations combine to form a cavitation cloud consisting of smaller bubbles, i.e., cavities. As the cavitation cloud reaches the surface, it becomes a ring vortex cavitation. Hence, the cavitation vortex collapses, thus hitting the surface in part of the ring. There are two developments of this method, cavitation jet in water and cavitation jet in the air [53]. Overall, the hydrodynamic cavitation can appear in a flowing liquid due to a decrease and subsequent increase in the local pressure-generating cavitation field [73,88].

#### 4.1.2. Acoustic Cavitation

Acoustic cavitation is called the growth and collapse of bubbles under the influence of an ultrasonic field [89]. Ultrasonic waves propagate in a liquid medium, causing mechanical vibrations of the liquid. Air bubbles contained in the liquid behave as specific cavitation nuclei, while areas of higher and lower particle density are created in the liquid [75,90]. As bubbles collapse, shock wave propagation and growth occur. This type of cavitation can occur under the influence of pressure field changes caused by oscillatory movements of the sonotrode excited by the transducer.

Apart from the benefits of cavitation action, it can cause material surface deterioration called cavitation erosion, resulting in erosive material loss. According to the authors’ previous papers, the erosion mechanisms for materials such as metals, plastics, and ceramics differ [91,92,93]. In the case of metallic materials, surface roughening resulting from plastic deformation and phase transformations usually precedes the fatigue-induced detachment of eroded material [94,95,96]. The test rigs employed for cavitation erosion testing primarily utilise hydrodynamic and vibratory cavitation. Vibratory cavitation erosion resistance tests are mainly conducted under ASTM G32 standard recommendations [97,98,99], while hydrodynamic cavitation test rigs usually conform to ASTM G134 standard (cavitating liquid jet) or are not-standard solutions, such as rotation discs or cavitation tunnel rigs [100,101,102]. Therefore, optimising the cavitation peening process parameters is necessary to prevent the surface from the erosive action of cavitation and obtain the required peening effects.

## 5. Shot Peening Effects

The main effects of SP but also CP include an increase in hardness, roughness, and residual compressive stresses. In addition, information is also given about the possible development of cracks, adverse effects on corrosion resistance, and others.

The effects of shot peening were investigated, among others, in a study made by Gonzales et al., which concerned the wear resistance of 18% chromium white cast irons after various heat and mechanical treatments to improve their properties [92]. The research carried out showed, among other things, that the best resistance characterises samples subjected to SP. The weight loss of the samples tested ranged from 2.583 to 1.819 mg/s, while after SP, it was 1.478 mg/s. It was also suggested that SP, in the case of the white cast iron mentioned, could be a more effective surface treatment than a long chipping treatment. The previous paper also presented the results of the percentage of residual austenite, which also favoured SP over other heat treatments. The content ranged from 11.65% to 6%, while after SP, it was 4.9%.

In the article by Bag et al., it was noted, among other things, that the development of cracks was slowed down in the zone affected by residual compressive stresses, and the distance between slowdowns for the samples after SP was equal to the austenite grain size [103]. A crack reduction was achieved for specimens between 3- and 5-grain size diameters, while the reduction of the stress amplitude from 1089 MPa to 931 MPa was twice as large as at a higher pressure. It was further concluded that the as-machined specimens’ average short crack growth rate could be well represented by the closure-free long crack growth rates.

One of the topics often addressed by researchers is residual stresses in a material. There is a lot of research on the effects of SP on material behaviour, and an interesting issue is also presented in the article [104]. The researchers analysed SP’s effects on duplex stainless steel (DSS) S32205, i.e., steel with a similar ferrite/austenite ratio. The steel under study was subjected to dissolution annealing at 1050 °C and water quenching. The effect of SP on the microstructure of the steel as well as on CRS (compressive residual stress) is described. It was concluded that SP has a greater effect on the microstructure and CRS of austenite than ferrite under the same SP conditions, however, in ferrite, main compressive residual stresses MCRS is present near the surface. In contrast, in austenite, high residual stresses are present near the surface, and MCRS is below the surface. The diameter of the nozzle for pressing was 15 mm, and the distance between the nozzle and the samples was 100 mm. The diffraction directions in the α phase and in the γ phase show that the SP process can change the microstructure in all directions of the diffraction plane. The higher micro-deformation, smaller domain size, higher failure probability, and higher dislocation density of γ-phase than α-phase in the near-surface layers are due to the higher work-hardening effect and lower SP influence on the microstructure [104].

The stress corrosion cracking of duplex steel after shot peening was described in the article [105]. The steel used in the study was 2205-Duplex steel, which was heat-treated at 982 °C for 30 min, cooled in the air for 2 h, and then quenched at 524 °C. The samples were then annealed at 529 °C. Shot peening was performed using steel balls. The results show that the time to failure increased 15 times for the best shot peening method. It was concluded that failure occurred after averages of 320 h and 25.5 h for the unpeened samples. At a stress level of 55%, failure did not occur after 335 h, and for unstressed specimens, failure averaged 5.6 h. Another conclusion is that the best peening results are obtained with the smallest possible particle size, which allows adequate peening without excessive wear on the peening balls.

Innovative research has also been carried out in the article [106]. Individual and hybrid methods of heat treatment, SP, and ultrasonic nanocrystalline surface modification USNM on cast aluminium alloy AlSi10Mg were analysed. The combination of these methods with each other and with heat treatment was also comprehensively examined, and the results of test studies were presented. The study showed the positive effect of the previously mentioned methods on the properties of the surface layer. Moreover, the best results were obtained for SP using low pressure and USNM with a high static load.

There have also been many papers describing the behaviour of different types of steel subjected to SP. Han X. et al. [107] reported the tribological behaviour of AISI 5160 steel subjected to quenching and tempering; moreover, the microstructure and residual compressive stresses were also investigated. The specimens were prepared by cutting 35 cm diameter and 10 mm high discs, then austenitised in a salt furnace followed by oil quenching and tempering. They were then hardened at temperatures ranging from 380 °C, 400 °C, 500 °C, 550 °C, 570 °C, to 590 °C. It was concluded that oxide wear particles formed on the surface of the disk, which behaved as a third body and adhered adhesively to the sphere, forming adhesive wear and micropitting on the sample surface. It was also concluded that shot peening and quench-tempering processes increase the residual compressive stresses, but SP has a better hardening effect on the soft disc. CRS can effectively increase the resistance to sliding wear and micro-flushing, but its effectiveness is more significant for hard discs that are hardened at low temperatures than for soft discs that are hardened at high temperatures. During tribotests, the tangential shear force and frictional heating generated by the frictional contact caused significant plastic deformation due to the ratchet process. The perlite/debonded martensite under the worn surface showed significant plastic deformation along the slip direction. The surfaces’ tempered martensitic/perlite microstructures showed mechanical fractures, fragmentations, and cracks. The corresponding ball samples had no detectable wear on the ball surface, indicating a lack of material transfer to the wear trace of the disc samples. Furthermore, the discs shot, peened, and hardened at 380 °C, 400 °C, and 500 °C showed minor wear compared to the non-peened coatings, but produced a larger wear volume. This was because the subsurface was too weak to support the upper surfaces during sliding.

R. Gopi et al. studied the friction and wear parameters of 316 L steel after shot peening with steel shots [108]. The study was conducted using the pin on the disc method. An improvement in morphology and resistance after shot peening was observed. The presence of shot peened and steel balls on the surface was observed. It was also observed that the roughness of the coatings after shot peening was lower compared to the values after spraying through the section. The hardness of the steel was expected to increase as a result of shot peening. The surface morphology was also improved without any negative effects on the formation of the secondary phase. It was also found that the addition of nickel has a beneficial effect on the stability of the austenite phase, even after shot peening.

Çakir et al. also tested austenitic steel, this time AISI 304 [109]. The samples were tested for corrosion and wear resistance with different coverage areas ranging from 100% to 1500%, and the effect of polishing after SP was examined. Wear resistance tests were carried out using the ball-on-disc method. The best wear resistance results were obtained at 1500% plating as the resistance increased by 70%, but crack initiation was noted. Significant improvements were also obtained at 100% and 200% coverage, where an increase of about 30% was obtained. Despite the increase in hardness, the corrosion resistance was not the best. It was also found that localised corrosion in chloride environments affected all samples. Polishing resulted in smaller pits, and the resistance was slightly higher, despite the lower passive layer formation potential.

M. Matsui H. Kakishima conducted studies on the improvement of steel properties under dry rolling, sliding wear with hard lube, and shot peening conditions [110]. The study showed improved properties under extreme sliding conditions. Prior shot peening treatment with ceramic balls produced a more uniform surface, and allowed the solid particles of the lube to remain on the surface.

H. Kovacı et al. presented a study on the hybrid treatment of ISI 4140 low alloy steel using pre-shot peening and plasma nitriding [111]. The specimens were subjected to different intensity peening at 16, 20, and 24 A, sequentially. Nitriding was carried out at 500 °C for 1 and 4 h. Wear tests were conducted under dry friction conditions, and the aim of the study was to characterise the effects of the dual process on friction and wear behaviour. It was concluded that both plasma nitriding and shot peening increased the roughness parameters and friction coefficient. It was also observed that the treatments increased the residual compressive stresses. An increase in wear resistance after plasma nitriding and SP nitriding was also found, but the efficiency of plasma nitriding was also improved. The untreated samples showed an adhesive type of wear and severe plastic deformation, while the treated samples showed more abrasive wear. The material, after plasma nitriding and shot peening, had a better bearing capacity.

Skoczylas et al. conducted an analysis of an Inconel 718 alloy subjected to pulsed SP using positron annihilation [112]. The roughness parameters increased, but also the hardness increased (by 25%). The average lifetime of positrons has also increased, which may be related, as suggested by the authors, to the increased annihilation of positrons in point defects.

Sliva et al. analysed the behaviour and properties of hardened ductile iron and ordinary ductile iron, without and after the SP process [113]. Hardened ductile iron is a cast iron obtained by hardening ordinary ductile iron. As stated by the developers, it has an interesting combination of favourable properties compared to steels of similar hardness, including toughness, wear resistance, fatigue resistance, and low cost. SP was found to lead to a transformation of residual austenite to martensite. It was also concluded that the increase in hardness was not sufficient to overcome the determining effect of coating roughness on wear behaviour, while the removal of the 20 μm layer allowed the positive surface hardening effects to be maintained. It can also be seen that the roughness, hardness, and coefficient of friction were highest for cast iron hardened after SP. The increase in hardness and roughness was greater for unhardened cast iron.

Żebrowski and Walczak conducted research on titanium alloy Ti-6Al-4V produced by the direct metal laser sintering (DMLS) method [114], and investigated various shot effects: chromium-nickel steel, crushed nut shells, and ceramic balls. Different working pressures were also used for SP 0.2 MPa, 0.3 MPa, and 0.5 MPa. The material consisted of gas-atomised powders of almost spherical shape, with requirements according to ASTM F1472 standard, and therefore consistent with the requirements for surgical applications for this alloy. Discontinuities of the structure formed were observed. Wear resistance tests were carried out using the ball on disc method with a force of 10 N in Ringer’s medium; the counter-sample was 6 mm in size and made of Al_2_O_3_. It is worth noting that DMLS technology itself makes residual stresses appear in the material [115]. The lowest wear resistance was found in the reference sample, i.e., the non-annealed sample, while the most robust wear test results were observed in the samples after SP using steel balls and ceramic beads at the highest pressure, i.e., 0.4 MPa. Furthermore, an improvement in toughness was observed at higher pressures for steel and ceramic beads. An increase in roughness was found for spherical beads, i.e., the previously mentioned ceramic and chromium-nickel steel beads. On the other hand, there was a decrease in roughness in comparison to the reference sample for the alloy after peening with nutshell particles. The values of friction and wear coefficients were lowest for steel shots and for ceramic beads, however, they increased with increasing pressure. The mechanism of material wear was adhesive.

Avcu et al. also conducted research on Ti-6Al-4V [116]. Two different peening times and two stainless steel sheet sizes were used. As a result of shot peening, hardness and roughness increased by approximately 35%. It was concluded that a longer than necessary peening time leads to the initiation of cracks, which has a direct negative effect on the abrasion resistance (reduction of 25%). It was also concluded that ball size affects roughness, with larger balls resulting in higher roughness values (0.74 µm for 0.09–0.14 mm diameter shots and 2.27 µm for 0.7–1.0 mm shots).

Walczak and Szala conducted tests on 17-4PH steel produced by a direct metal laser sintering DMLS method [117]. The sintering was carried out according to the manufacturer’s recommendations in a nitrogen shield, and discs of 30 mm diameter and 6 mm height were produced. Tests were carried out on a reference sample and using a CrNi steel shot, nutshell granules, and ceramic beads at different pressures of 0.3 and 0.6 MPa. Peening time was 60 s and standoff distance from the nozzle was 20 mm. The average grain size for stainless steel ranged from 400 μm to 900 μm, 450–800 μm for nutshells, and 125 μm to 250 μm for ceramic beads. Roughness, microhardness, morphology using SEM, wear resistance using the ball-on-disc method, and corrosion resistance in 0.9% NaCl solution for the steel after SP were investigated. XRD was also used to confirm the presence of martensite and residual austenite in the alloy structure. The use of peening except for nutshells resulted in the formation of martensite and a reduction in residual austenite. An increase in roughness was observed after dressing in steel balls and walnut shells, and the roughness values increased with increasing dressing pressure, but the opposite trend was observed after dressing with ceramic balls. An increase in surface hardness was also observed with each of the pressing media. The greatest increase in hardness was recorded for the specimens annealed at 0.6 MPa with ceramic and steel balls, 119% and 116%, respectively. An improvement was also observed, i.e., a decrease in the consumption coefficient K and an increase in the friction coefficient, and these changes occurred with increasing pressure, so that with increasing pressure, the coefficient K decreased, but the friction coefficient increased. The best resistance was found in the material after pressing with ceramic balls in corrosion tests, followed by CrNi stainless steel arrows. The worst results were obtained after peening with walnut shells. In addition, it was found that, overall, the best results were obtained with ceramic balls at a pressure of 0.6 MPa. To summarise, Table 2 shows some of the results that have been achieved in shot peening studies.

## 6. Cavitation Peening Effects

The first research on ultrasonic material processing is attributed to Takahashi et al. In 1987, and they presented a paper in which it was observed that the fatigue limit increased by about 11% under the same conditions, i.e., 0.5 mm sonotrode distance from the sample surface and 15 kHz frequency [124]. It was also concluded that the method could be used to perform penetrations in the material and to produce patterns following the shape of the sonotrode.

Cavitation peening was analysed, among others, by Y Gao et al. [125]. The hardness at different depths from the surface, the roughness, the length of the peening, and the different vibration amplitudes of the sonotrode from 20 to 100%, with a maximum vibration amplitude of 20 μm, were analysed. The test materials were 304 stainless steel and 200 nickel alloy. The test rig consisted of a water bath filled with distilled water in which a titanium sonotrode horn with a 19 mm tip diameter was placed, vibrated at 20 kHz by a converter powered by a generator and controller. The clearance between the sonotrode tip and the workpiece was 1 mm. The Vickers hardness for 304 steel increased from about 230 HV at zero amplitude to 260 HV at 20 μm, with a processing time of 10 min. For the nickel alloy, the hardness increased from approx. 115 HV at zero amplitude to 170 HV at 20 μm amplitude; the processing time was 3 min. For both steel and nickel alloys, the clear point after which the increase in hardness occurred was 10 μm. An increase in hardness was already recorded at 6 μm. In the case of 304 steel, a hardness increase was recorded with a maximum amplitude of about 18% and 36% in the case of the nickel sample. A roughness increase was noted with increasing amplitude, but this was at a fairly low level, even at an amplitude of 20 μm. The roughness increase was probably due to an intensifying of the cavitation process due to the increase in amplitude.

Research regarding the strengthening of aluminium by cavitation peening was also conducted by H Soyama et al. [126]. They used a JIS AC4CH aluminium alloy, which is similar to ASTM A356.0. However, the method of cavitation excitation was different from that of Gao. An underwater cavitation water jet was used to produce cavitation strokes. The nozzle was cylindrical and had a diameter of 2 mm with a length of 6 mm. To show the fatigue strength, the researchers conducted rotary bending fatigue tests. The fatigue strength improved by 50% compared to the uncavitated part, at relatively low pressure. It was concluded that cavitation pressing produces different impact intensities as well as pits of different sizes due to plastic deformation. It was suggested that this method could be used as an alternative to the single or multiple shot peening of surfaces.

Ye et al. performed a study on the AZ31B magnesium alloy [127]. Two media water and kerosene were used. In addition, different stand-off distances, amplitudes, and treatment times were applied. The highest hardness was obtained in water at an amplitude of 75% distance of 1.0 mm and a time of 20 min, and the smallest grain size for this sample of 2.014 μm was also detected. The hardness after treatment in water was 3.77–48.19% higher than in kerosene. It was also concluded that grain fineness is key to increasing hardness, and the size after treatment in kerosene was larger than in water, from 1.667 to 4.479 times.

D.Y. Ju and B. Han [63] conducted cavitation peening tests for technically pure titanium. Specimens A and B were placed stationary in a clamp and subjected to aqueous cavitation pressing. Tests were carried out for 15, 30, 45, and 60 min of cavitation peening. It was also suggested to divide the CRS into two main levels of macro and micro stress. The study showed that cavitation peening can induce significant residual stresses, with depths of up to about 150 μm. It was also concluded that the dislocation density of the pure titanium phase-α increases when the impact energy caused by water-induced cavitation annealing exceeds the internal energies of the pure aluminium crystal in the early stages of annealing. Twinning-induced deformation, the interaction between these deformed twins and dislocations, and local microplastic deformation are induced by water cavitation peening (WCP) in the strengthening layer. These results indicate that the mechanism of WCP amplification may be related to both the activity of twin deformation and dislocations.

Fukuda S. et al. [128] conducted a study regarding the fatigue resistance improvement of steel after nitrocarburization. JISS50C steel was tested after cavitation peening with an underwater water jet, and in a non-corroded condition. Hardness, roughness and residual compressive stresses were investigated. The material was hardened and tempered to approximately 290 HV, then fatigue test specimens were nitrocarburised. After being nitrocarburised, the specimens were oil-quenched. As a result of cavitation peening, the maximum hardness in the diffusion zone increased from 377 HV in the case of an unburnished surface to 413 HV. The maximum residual compressive stress increased from 305 MPa to 434 MPa. The fatigue strength limit was 668 MPa for the non-welded samples and 772 MPa after cavitation peening. It was concluded that the roughness was not significantly affected with an optimized arc height of 0.15 mm/N (increase from 0.24 μm to 0.28 μm). It was also found that it was possible to estimate the increase in fatigue limit by measuring hardness and residual compressive stresses.

Numerous studies on the fatigue life of components such as pinion shafts and rollers were carried out by Seki and Soyama [129]. Originally, an attempt was made to use CP technologies for gear shafts; because SP for peening is a standard method, it was decided to use CP. The test pieces of equipment used in this study were gear shafts and rollers; two types of chromium-nickel steel SCM 440, SCM 415, and a counterexample in the form of SCM 420 steel were used. The samples were cut into 60 mm diameter discs, ground and heat treated. Hardness, roughness and residual stresses were investigated. It was found that CP increased the residual compressive stresses (RCS) and hardness of the gear shafts. It was noted that the roughness was at a similar level after CP as non-peened. The failure mode of the heat-treated test rolls was pitting due to cracking, while the surface-treated test rolls were spalling due to subsurface cracking. In the case of the toothed roller test, there was spalling resulting from surface cracking. Fatigue strength improved as a result of increased hardness, as well as an improvement in surface fracture toughness for the pinion rollers.

Seki et al. conducted a study to determine the rolling fatigue life of gears and rollers [130]. For this purpose, gears and rollers and mating rollers and rollers were prepared. Specimens with a diameter of 60 mm were cut out. Gear rolls and rollers were made of JIS: SCr420 chromium steel and mating rollers and gear wheels of JIS: SCM 420. The test pieces were surface hardened and ground. In addition, two CP cases were used in the tests, using a water jet with low (0.2 MPa) and high injection pressure (30 MPa). The tests were carried out successively at 1, 3, and 5 min. The lubrication used was ATF oil for the rollers and gear oil for the gear pairs. The results were as follows: CP increased the hardness and residual compressive stresses in the test specimens, but in the case of the gears, the increase in hardness was not significant. Wear marks were similar for both annealed and non-annealed samples. The fatigue strength of the samples after CP was similar for the samples after SP.

Seki et al. also investigated the behaviour of steel after rolling contact fatigue [131]. The material after SP, fine particle peening (FPP), and after cavitation peening (CP) and a reference (NP) sample were investigated. The paper also presents a summary of parameters and properties for the samples mentioned above, including hardness, roughness, and residual compressive stress. Roughness is essential for the life of a component under rolling contact fatigue conditions; an increase in roughness generally leads to a decrease in strength.

Research on the application of cavitation has also been carried out by Fatyukhin et al. [132]. Their study mainly analysed the long-term (10 h) effect of ultrasonically induced cavitation at a sonotrode distance of 4 mm from the test material, which is longer than most researchers. Two grades of steel were used in the study, namely 45 and 40 Kh. It is worth noting that the paper also presents changes in the morphology and roughness of the tested steel under cavitation conditions. Preliminary tests showed the optimum measurement moments at 10, 20, 30, 40, and 60, and tests were carried out every 1 h. For 40 Kh steel, the first signs of erosion were observed after 20 min of testing, as was the case for 45, despite the similar composition and structure (the main difference being the chromium content of 40 Kh at about 1% and 0.1% for 45, with a slightly higher perlite content). The tests showed that the hardness for 45 increased after 20 min and then began to decrease, while the hardness for 40 Kh steel did not increase compared to the untreated sample, but decreased. The potential to change the roughness of complex shaped objects and cavitation for cleaning was suggested.

He et al. investigated in a TC4 titanium alloy subjected to cavitation shot peening [133]. Tests were carried out on five samples subjected to different pressures, times and nozzle diameters. Roughness, microhardness, metallographic tests as well as residual stresses and phase composition analysis were carried out. The depth of the influence of cavitation peening on microhardness and residual stresses at up to 130 μm below the surface were determined. The formation of deformation twins emerging in some of the coarse grains of the α-phase was also detected. The depth of the deformed zone was determined to be 55–60 μm. An increase in hardness of up to 33.6% was also observed.

A new approach to improving the ultrasonic cavitation peening (UCP) pressing process was proposed by Fushi Bai et al. [134]. They proposed to introduce a water jet into the gap between the sonotrode and the workpiece to be burnished, which removes the ASTM-compliant water tank and temperature control system from the bench. The results for hardness, roughness, and volume loss were similar to those obtained with the ultrasonic method, the authors said. By using this method, it was possible to obtain a larger spacing, and moreover, the stand was simplified.

Gu et al. [135] used laser cavitation to improve the structural integrity of grey cast iron. They investigated two materials; HT250 grey cast iron and HT250 grey cast iron with a 0.1 mm thick copper layer. They also simulated the collapse of a cavitation bubble, presented the velocity field distribution for a bubble imploding near the surface, and presented the stress-strain field distribution in the material and the copper layer. The maximum microbubble velocity was 380 m/s, and the diameter was 0.1 mm. Morphologies and laser parameters’ effects on roughness, microhardness, and residual compressive stress were analysed. The most optimal microhardness and residual compressive stress results, and the lowest roughness, were obtained with a laser energy of 200 mJ. The optimum values of roughness and microhardness were obtained at a defocusing amount equal to 0 mm for the sample with the copper layer, while for the material without the layer, it was 1 mm. It was found that the use of a copper layer allowed a more homogeneous plastic deformation while reducing the roughness after laser cavitation shot peening by half compared to an uncoated surface.

The dynamics of laser-induced cavitation bombardment were investigated by Burjan et al. [136]. The study was conducted using an Nd: YAG laser, and the body in the vicinity of which the cavitation was induced was a polyacrylamide gel with a water concentration of 80%. Images of the jet were taken with a camera at up to 5 million frames per second. The maximum bubble diameter was determined to be 1.55 ± 0.05 mm, the maximum velocity of the microburst was 960 m/s, and in the opposite direction, 600 m/s.

Cavitation peening can improve the surface properties of material similarly to SP, although the principle of the two methods is slightly different. Both the results for depth of stress profile, CRS, and surface parameters such as hardness and roughness are different, depending on the material, method, and process parameters used. Comparing stress profiles, it can be concluded that SP is able to generate higher CRS close to the surface, whereas after cavitation peening, slightly higher stresses can be encountered deep in the material, and their distribution is more even/mild, while higher surface hardness is obtained after SP. The selection of the peening method should be based on the material, the location, and shape of the component, the corrosion resistance requirements, or the required surface profile [137,138]. Moreover, based on the literature review, it is concluded that a combination of both cavitation and shot peening processes could provide promising results of surface treatment, which unfortunately has not been described by the literature. The combination of CP and SP methods was suggested in connection with the search for new solutions in the field of shot peening. This is because different materials respond differently to shot peening, especially after heat treatment, thermochemical treatment, and the use of different manufacturing or processing methods. It is worth considering a combination of SP and CP, as it can have positive effects in terms of the depth of the layer strengthened and the quality of the surface, as well as giving greater control over the strengthening effect. A combination of these methods can also be useful due to the surface cleaning effect of cavitation, which can have positive effects on the reduction of cytotoxicity in biomaterials, and reduce the corrosion potential. To summarise, Table 3 shows you some of the results that have been achieved by the cavitation peening research

## 7. Modification of Components Produced by Additive Technologies

Additively manufactured (AM) components made of ferrous and non-ferrous materials are systematically studied. AM processes for metals are typically divided into two groups; first is directed energy deposition (DED) processes like laser metal deposition LMD [38,143], the second is powder bed fusion (PBF) methods like selective laser melting (SLS) and direct laser metal sintering (DMLS) [101,102,103]. Thanks to them, it is possible to produce objects of an even more complex shape, but the material after printing requires additional processing to improve its properties. Both cavitation and shot peening methods can be used to improve material properties after applying additive manufacturing techniques. Figure 6 shows the categorisation of the surface post-treatments applied to AM metallic materials.

Soyama and Sanders investigated a material produced by the electron beam powder bed melting (EBPB) printing of Ti6Al4V powder [54]. Samples with a thickness of approximately 2 mm were used in the study. The samples were heat-treated in a vacuum at 936 °C for 105 min. The specimens were then also aged in a vacuum at 706 °C and subjected to cavitation burnishing. The method used for pressing was a cavitating jet in water. The study’s authors concluded that, with the optimal process parameters, an improvement of 66% in fatigue strength was achieved, which was due to a combination of heat treatment, the introduction of CRS, and largely through the reduction of roughness and strengthening crushing. In addition, the introduction of CRS accounted for 8% of the total contribution to improving fatigue life.

Tan and Yeo conducted a study in which they used a combined surface treatment after DMLS [140]. The study used Inconel 625 alloy; two test specimens were produced by cutting from a plate, one cubic and the other also cubic with three 3 mm diameter channels. This choice of specimens was intended to best represent the shape of the parts produced using this method. The parts were then aged at 870 °C for 1 h. Additionally, ultrasonic cleaning was carried out to remove loose powder particles. Cavitation was induced using a vibratory apparatus, and the treatment time was 45 min. Investigations were performed on an as-built surface after pure cavitation and Ultrasonic Cavitation Abrasive Finishing (UCAF). Tests were carried out on different component building orientations 0, 45, 90. Figure 7 shows the mechanism of particle removal from the surface produced by means of powder bed fusion (PBF) under cavitation conditions.

The investigations showed that ultrasonic treatment combined with abrasion could improve surface properties after DMLS. The entire surface had a uniform appearance and Ra between 2.7 and 3.8, with an increase in hardness of 15%. However, the most important finding is a 20% improvement in Ra inside the holes. Although this was done for 3 mm diameter holes and required further study, it is a promising application for cavitation peening. In addition, it was found that increasing the gap from 0.813 mm to 2.007 mm had a negative effect, i.e., decreasing Ra from 40% to 28%.

Soyama and Okura compared properties for Ti6Al4V titanium alloy after 3D printing by electron beam melting (EBM) [145]. The test specimens were subjected to CP, SP, and LSP, and compared with a non-cavitation peened reference specimen. Cavitation peening was done by injecting a high-pressure water jet into the water. SP was performed using a water jet to accelerate the particles. Increased fatigue strength after SP, CP, and LSP of 68%, 84%, and 104%, respectively, were obtained. The roughness parameters Ra for the CP, LSP, and unstressed samples were similar and within standard deviation, but significantly decreased for the SP sample. Similarly, in the case of the Rz parameter, i.e., the maximum height of the roughness profile, the LSP sample was an exception as it obtained a slightly higher value; as the authors suggest, this was due to the nature of the process, where larger and larger valleys were formed in each pass. Hardness was examined using the HR15N and HR15T methods. The work hardening effect was investigated, and the hardness was measured using the Vickers method; it was observed that the hardness after CP and SP increased to 367 ± 17 HV and 386 ± 26 HV, respectively. The unglued sample had a hardness of 344 ± 6 HV, and in the case of LSP, there was a softening of the material to 338 ± 21 HV, which may be related to the large amount of heat generated during ablation.

Żebrowski et al. described the mechanical properties and cytotoxic behaviour of titanium alloy Ti-6Al-4V produced by DMLS technology [146]. Gas-atomised Ti-6Al-4V powder was used in the study. After fabrication, annealing heat treatment was applied for 4 h at 800 °C in a vacuum and cooled in argon. The samples were then subjected to SP at an operating pressure of 0.4 MPa to obtain 100% coverage. Increases in hardness of about 42% and 30% were obtained after the peening of CrNi steel shots and ceramic balls, respectively. In Figure 8, the cross-section of the material with a visible hardened layer is also shown.

The relationship between the peening medium used and the height of the reinforced layer is apparent. Steel ball shot peening provided a large depth for the strengthened zone, whereas ceramic balls and walnut shells provided the successively reduced depth of the strengthened layer. The depth of consolidation can be attributed to the energy delivered by a single shot.

There was a clear correlation between the height of the reinforced layer, the hardness, and the tensile strength limit. The best results were unmodified < nut shells < ceramic beads < CrNi steel shots. Cytotoxicity was lowest for ceramic balls and then increased for nut shells. The highest cytotoxicity was obtained for steel shots. With respect to mechanical properties, the best results were obtained with steel shots, whereas with respect to biocompatibility, better results were obtained with ceramic balls.

Uzan et al. investigated the effects of SP after sintering by the SML method, and the results of fatigue life were compared to die casting from the same material, namely AlSi10Mg [147]. The hardnesses of the as-cast and SML-produced samples were 87 HV and 95 HV, respectively. It was concluded that optical–mechanical and electrochemical polishing with removal of about 25–30 μm leads to a significant improvement in fatigue resistance and a significant reduction in roughness. Differences were also noted in the type of cracks that occurred for the part after additive manufacturing as well as for the die-cast part. The after AM part was ductile with relatively deep indentations, while the die cast part was more brittle, and contained numerous microcrack facets. Table 4 shows the percentage change obtained in different peening treatments. It is worth noting that the results cannot be directly compared with each other, as most were obtained using different peening methods, and with different parameters. Consequently, Table 4 outlines what can be achieved using the different methods.

## 8. Summary

This paper reviews the basic information regarding shot peening methods and the effects of the shot peening (SP) and cavitation peening (CP) treatments on the surface layer properties of metallic components. In addition, the literature survey done regarding the cavitation and shot peening treatments allows one to state the research gaps and further directions of the surface treatment process of metallic structures.

There is a research gap since, as far as the authors’ knowledge goes, there are no studies describing a hybrid treatment consisting of a combination of shot peening and cavitation peening methods. A research area in this direction could include, for example, the application of CP treatment after SP, or CP treatment followed by SP. In addition, the optimisation of hybrid process parameters such as different peening times, intensities or shot sizes in the SP process is also required.

There are few studies on the cavitation treatment of materials after additive manufacturing of metallic structures. Most papers are limited to reporting the cavitation treatment of conventional metallic materials, as most of them concern steels, aluminium and titanium alloys.

The SP and CP effects for materials obtained by 3D additive manufacturing are similar to or even better than the materials produced by classical manufacturing methods. The 3D printing surfaces can be inhomogeneous in their structures, which can be eliminated by peening. In addition, reducing the number of nonuniformities can positively affect corrosion resistance.

The results obtained by the peening of additive technologies may differ depending on the method used for printing and the parameters or gas used in the shot peening process and employed peening method. Increasing the intensity and parameters of the peening process does not always lead to positive results. Roughness and structural integrity can suffer in particular.

The literature gap leading to the scope of future work seems a need to investigate the effects of peening, especially cavitation peening and hybrid peening, on the anti-wear and corrosion performance of additively manufactured metallic materials.

## Figures and Tables

**Figure 1 materials-15-02476-f001:**
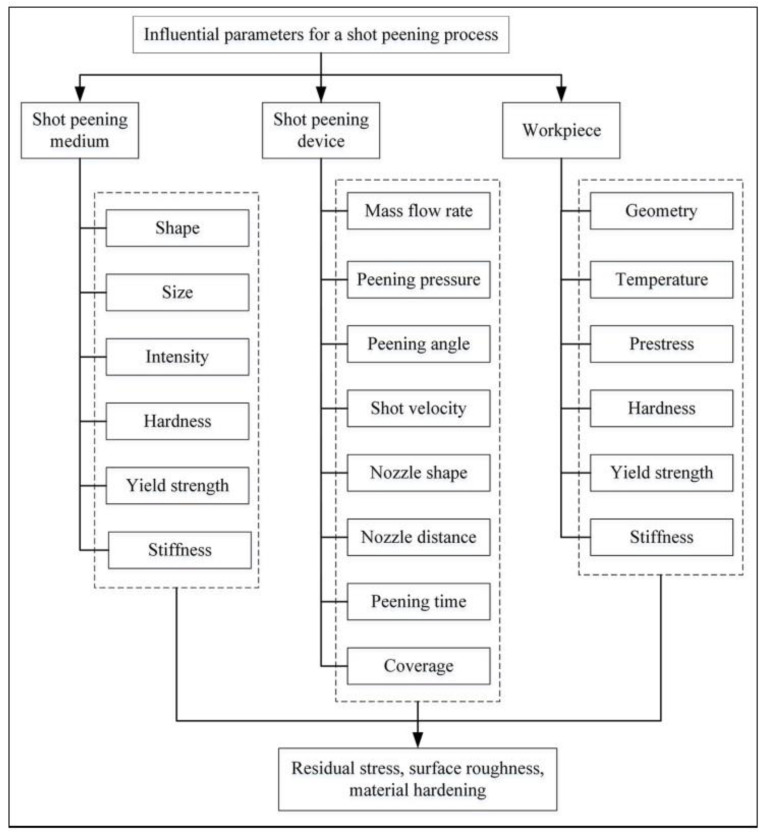
Influential parameters and products of SP [29].

**Figure 2 materials-15-02476-f002:**
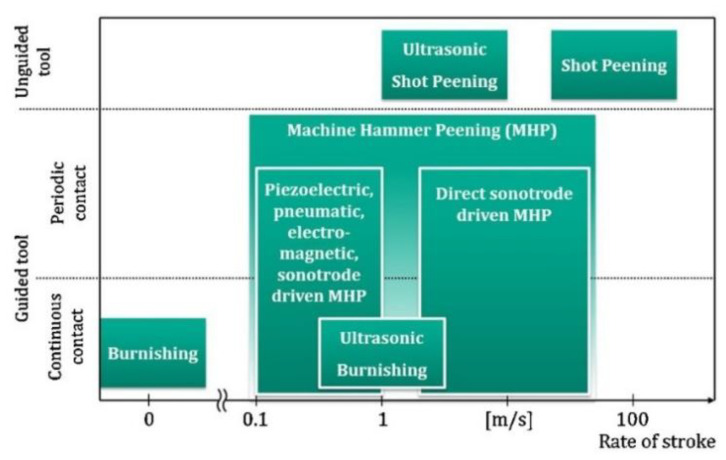
Overview of surface modification processes with guided and unguided tools (reproduced with permission from Ref. [30], Copyright 2016, Elsevier).

**Figure 3 materials-15-02476-f003:**
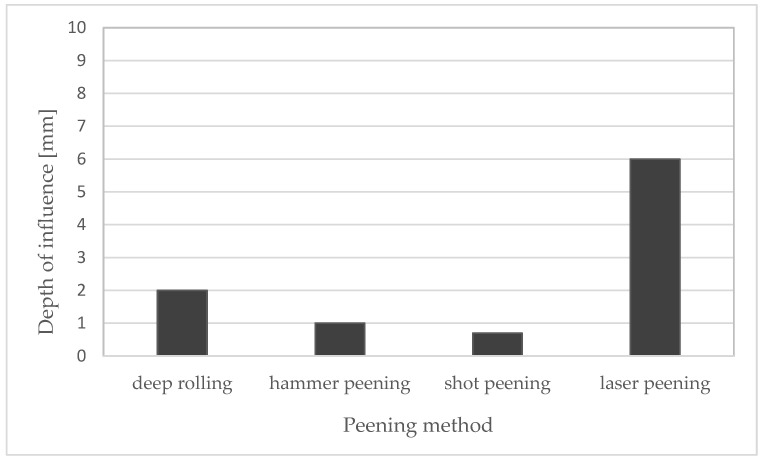
Depth of influence of deep rolling, and hammer peening, shot peening and laser shock peening—comparison. Compiled on the basis of [34].

**Figure 4 materials-15-02476-f004:**
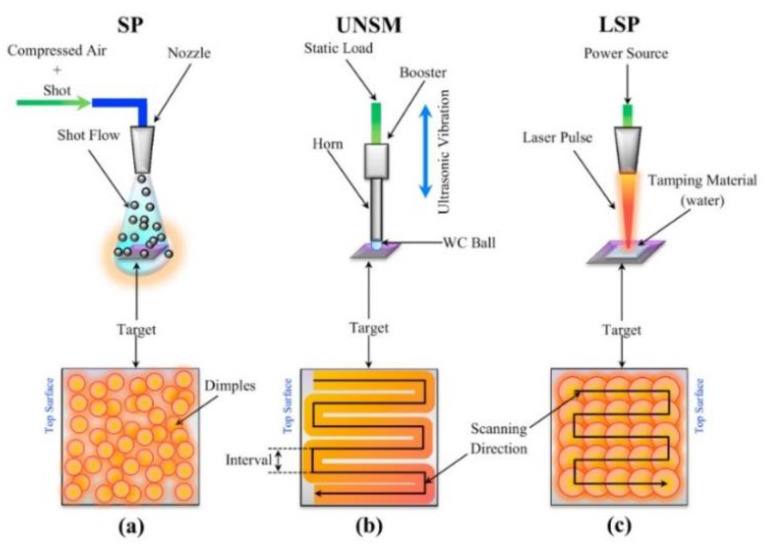
Schematic illustration of different surface SPD treatments and the corresponding plastically de-formed top surface of the target material: (**a**) SP, (**b**) UNSM, and (**c**) LSP processes (reproduced with permission from Ref. [46], Copyright 2021, Elsevier).

**Figure 5 materials-15-02476-f005:**
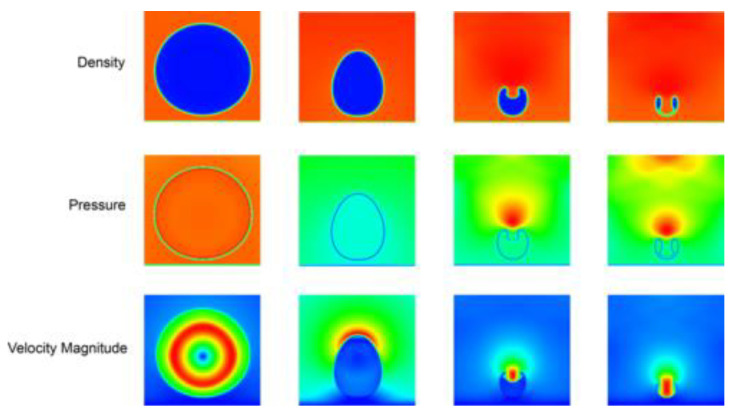
Sequences of collapse process of a H_2_O vapour bubble near a solid wall (from **left** to **right**), shown by density, pressure, and velocity magnitude fields (reproduced with permission from Ref. [82], Copyright 2020, American Physical Society).

**Figure 6 materials-15-02476-f006:**
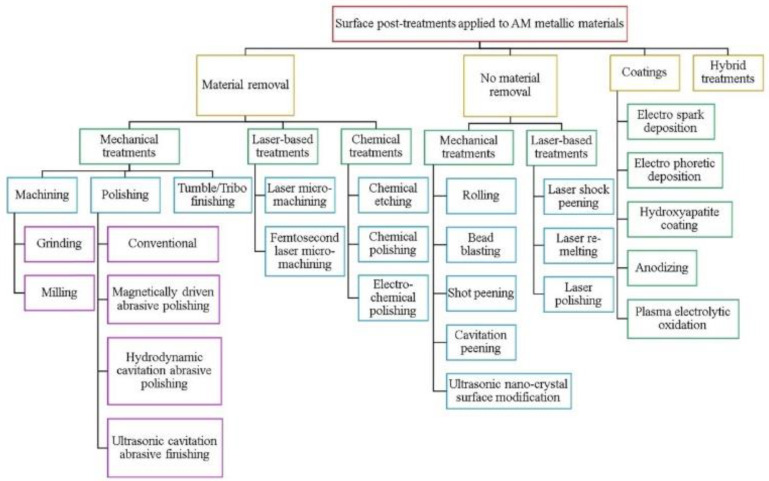
Categorisation of the surface post-treatments applied to AM metallic materials (reproduced with permission from Ref. [144], Copyright 2021, Elsevier).

**Figure 7 materials-15-02476-f007:**
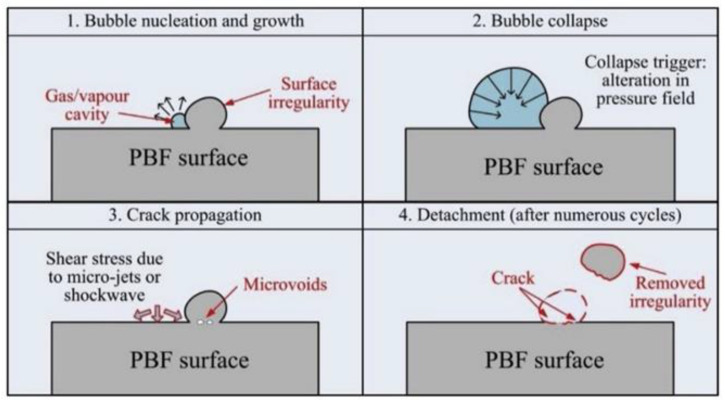
Heterogeneous cavitation on PBF surfaces (reproduced with permission from Ref. [140], Copyright 2020, Elsevier).

**Figure 8 materials-15-02476-f008:**
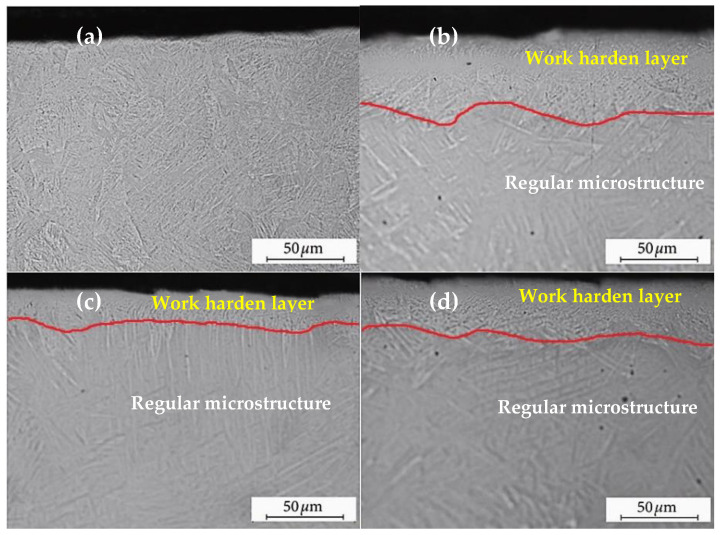
The cross-section of specimens showing modified surface layer after shot peening using the pressure of 0.4 MPa: (**a**) unmodified surface after DMLS (reference sample); (**b**) shot made of CrNi steel; (**c**) nutshell granules; (**d**) ceramic beads, author’s elaboration based on [146].

**Table 1 materials-15-02476-t001:** Search results for the phrases: cavitation peening (CP), shot peening (SP), and cavitation peening plus shot peening (CP + SP).

Total Number of Results	Shot Peening	Cavitation Peening	Shot Peening and Cavitation Peening
WoS	3824	200	82
Scopus	4807	194	90

**Table 2 materials-15-02476-t002:** Effects of SP on materials’ properties and performance.

Material	Technique	Findings	References
JIS SUS316L	SP- Particles accelerated by water jet	Enhance the fatigue strength by 25%	[118]
M50	USP	The maximum hardness increased by 24% -The wear rate decreased by 50.4% under sliding conditions	[119]
17-4PH	SP	Hardness increased by 30–52% Roughness decrees by up to ~70%	[120]
AISI 4140	SP	SP treatment increased the corrosion resistance of the material	[111]
Al 7075	SP	The highest microhardness 50 µm below the surface was found for the in quenched condition (Q) and the samples quenched and aged at 145 °C (Q-145), while the lowest was found for the samples quenched and aged at 195 °C (Q-195).	[121]
Forged Ti-834	SP	The density of mechanical twins increases with increasing peening coverage, but increasing peening coverage does not increase the depth to which mechanical twinning occurs.	[122]
AISI 304	wet shot peening	Increased maximum microhardness by 64% for 100% coverage and 88.16% for 500% coverage compared to the base metal.	[123]

**Table 3 materials-15-02476-t003:** Effects of CP on materials properties and performance.

Material Type	Technique	Findings	References
Magnesium alloy AZ80	CP	Micro hardness in the surface layer after cavitation peening increased for about 20–40%	[121]
JIS A2017-T3	WJP	Cavitation peening increased the the fatigue life of a duralumin plate with a hole with a chamfered edge by 286% and by 1100% for the specimen with a round-edged hole	[139]
Inconel 625	Ultrasonic cavitation abrasive finishing	Slight surface hardening of up to 15% The 20% Ra improvement on the internal Surfaces	[140]
TC4 Titanium	WCP	The depth of influence of cavitation peening on microhardness and residual stresses was determined to be up to 130 μm. An increase in hardness of up to 33.6% was also observed.	[133]
304 Ni-200	UCP	Increase in hardness by 32%, increase in fatigue life by 400% Increase in hardness by 18%, increase in fatigue life 833%	[125]
EN 10088-3 EN AW-2030 T3	UCP	Stainless steel surface hardness increased more than two times, while aluminum only by 14%	[141]
5456-H116 5083-H321	UCP	The lowest corrosion current density was found at 3.5 min for 5456-H116 and 4.0 min for 5083-H321	[142]

**Table 4 materials-15-02476-t004:** Comparison of changes in hardness, roughness (Ra parameter) and fatigue life reported for popularand additive manufactured metallic materials. The values in the table are the percentage increase or decrease [%] due to peening process.

Popular Metal Alloys			Additive Manufactured Parts		
Hardness	Roughness	Fatigue Life	Hardness	Roughness	Fatigue Life
+14	[107]		+13	[106]	
−3			+7		
−1			+54		
+1			+74		
−1			+71		
+3			+83		
+28			+62		
	[124]	+11	+77		
+18	+833	[125]	+106	+15	[117]
+32	+400		+116	+18	
+11	[63]		+10	+28	
+10	+17	+15	+27	+43	
	[128]		+108	−7	SP
+2	+9	[130]	+119	−15	
+4	+50		+36	[134]	CP
+8	+200	Similar to	+(14–18)	Up to 800	+66
+1	−10	SP		[54]	
+1	−3		Up to 15%	−30	[140]
+6	-			−20 in holes	
+17 CP	+107	[131]	+12	No change	+68 SP
+ 54	+154		+6	[145]	+84 CP
+28	+115		−2		+104 LSP

## Data Availability

No new data were created or analyzed in this study. Data sharing is not applicable to this article.

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
