# Peer review of "Effects of Shot Peening and Cavitation Peening on Properties of Surface Layer of Metallic Materials—A Short Review"

_materials, 2022, doi:10.3390/ma15072476_

Round 1

Reviewer 1 Report

The present manuscript is an interesting review paper about shot peening and cavitation peening. The presentation and organization of this manuscript are not very good. The sections of “6. Shot peening effect” and “7. Cavitation peening effect” should be reviewed more in detail. The authors just summarized and reviewed in order of the literature in these sections. They should be organized better way such as the classification of each effect on microstructure, surface roughness, and so on. 
Additionally, there is some mistake in the numbering of sections. Please check it.

Author Response

Dear Reviewer #1,
We wish to thank the Reviewer for acknowledging our work and recommending it for publication. We also wish to express our gratitude for all comments on how to improve the manuscript. We have taken all suggestions into account and made necessary revisions, as explained below. The revised fragments are marked in yellow. We hope that the manuscript is now more readable and coherent.
The main revisions made in the manuscript and our responses to the Reviewer’s comments are as follows:
Reviewer #1: The present manuscript is an interesting review paper about shot peening and cavitation
peening. The presentation and organization of this manuscript are not very good. The sections of “6.
Shot peening effect” and “7.
Response: We have made the following changes to improve the organization of the manuscript:
Reviewer #1: Cavitation peening effect” should be reviewed more in detail.
Response: Following your remark and reviewer #3 comment, we have expanded this section and
added tables with the effects of cavitation peening and shot peening.
Reviewer #1: The authors just summarized and reviewed in order of the literature in these sections.
They should be organized better way, such as the classification of each effect on microstructure,
surface roughness, and so on.
Response: We have added tables including the effects of cavitation peening and shot peening on
material properties such as microstructure, surface roughness, and so on.
Reviewer #1: Additionally, there is some mistake in the numbering of sections. Please check it.
Response: We have corrected the numbering mistake and eliminated the other typos.
Thank you for considering our manuscript for publication.
Yours sincerely,
Aleksander Świetlicki and Mirosław Szala
(corresponding authors)

Reviewer 2 Report

Review must be current. The number of cited publications is sufficient, but 29 of them are older than 15 years, ie 27 %.
Publications [74], [75] must be cited by standard.
Group citations can be used especially in Review to a limited extent, for a maximum - 3 citations (see 10-14, 68-72, 99-103).

„Moreover, on the basis of the literature review, it is concluded that a combination of both cavitation and shot peeening processes could pro vide promising results of surface treatment, which unfortunately has not been described by the literature.“
The reader will ask: "why?". What benefit do we want to achieve in technical practice by combining these methods? These methods are applied differently. Of course, both methods involve plastic deformation and increase of compressive stress in surface. In the case of the cavitation method, the depth of the affected layer is much smaller, it is preferably the non-destructive character of the new surface (without microcracks). Contrary, shot peening is a relatively more destructive method, but more effective for hardening of the surface.

Author Response

Dear Reviewer #2,
We wish to thank the Reviewer for acknowledging our work and recommending it for publication. We also wish to express our gratitude for all comments on how to improve the manuscript. We have taken all suggestions into account and made necessary revisions, as explained below. The revised fragments
are marked in yellow. We hope that the manuscript is now more readable and coherent.
The main revisions made in the manuscript and our responses to the Reviewer’s comments are as follows:
Reviewer #2: Review must be current. The number of cited publications is sufficient, but 29 of them
are older than 15 years, ie 27 %.
Response: According to your remark, we modified the references list, added some current paper and
deleted some of the old publications. Therefore, we reduced the percentage of references older than
15 years to 19.73 %. Thank you for this remark.
Reviewer #2: Publications [74], [75] must be cited by standard.
Response: Thank you for that remark. In the whole manuscript, the referencing style has been unified
according to the MDPI standards
Reviewer #2: Group citations can be used especially in Review to a limited extent, for a maximum - 3
citations (see 10-14, 68-72, 99-103).
Response: Group citations have been corrected as follows so that the number does not exceed 3.
Reviewer #2: „Moreover, on the basis of the literature review, it is concluded that a combination of
both cavitation and shot peeening processes could provide promising results of surface treatment,
which unfortunately has not been described by the literature.“
The reader will ask: "why?". What benefit do we want to achieve in technical practice by combining
these methods? These methods are applied differently. Of course, both methods involve plastic
deformation and increase of compressive stress in surface. In the case of the cavitation method, the
depth of the affected layer is much smaller, it is preferably the non-destructive character of the new
surface (without microcracks). Contrary, shot peening is a relatively more destructive method, but
more effective for hardening of the surface.
Response: The combination of CP and SP methods was suggested in connection with the search for
new solutions in the field of shot peening. Because different materials respond differently to shot
peening, especially after heat treatment, thermochemical treatment, and the use of different
manufacturing or processing methods. It is worth considering a combination of SP and CP, as it can
have positive effects in terms of the depth of the layer strengthened and the quality of the surface, as
well as giving greater control over the strengthening effect. A combination of these methods can also
be useful due to the surface cleaning effect of cavitation, which can have positive effects on the
reduction of cytotoxicity in biomaterials, and reduce the corrosion potential.
Thank you for considering our manuscript for publication.
Yours sincerely,
Aleksander Świetlicki and Mirosław Szala
(corresponding authors)

Reviewer 3 Report

  1. Please mention the following statement in the Abstract at the end as “Literature gap leading to scope of future work is also included.”
  2. Introduction, 2nd para seems to be like writing for text book, without any literature support. As justifications, please add some literature support for the arguments.
  3. Page 2, last sentence, “The effects of these treatments are in this work are described and the results are presented after 3D printing of metallic components”. Please correct the grammatical mistake.
  4. Page 3, 1st paragraph, “For this reason, various peening methods have been used for machining parts made of metallic materials produced by printing and combinations of peening methods with other machining methods, such as heat treatment, are also used.” There is ambiguity in the sentence. How heat treatment is the machining method?
  5. I could not understand about how the figure 1 (Unary phase diagram of water) is fitted to the present discussion? The phase transformation process is related to Gibb’s phase rule, where the number of phases at equilibrium is connected to the variables and degrees of freedom. Please explain. Otherwise remove figure 2.
  6. A table, containing the types of cavitation, effect, benefits, type of surface damage quoting published papers may be useful to the users. This can be added in the manuscript.
  7. Similarly, under shot peening, a table, containing the development of surface roughness, hardness, corrosion resistance, fatigue resistance and fretting as per different works in the domain by different researchers can be included to add value to the manuscript.
  8. Similarly, shot peening effects may be presented in the form of table, comprising of materials, process parameters, property improved etc. to minimise monotonous approach of writing as well as more data may be show cased.
  9. Figure 10, no clarity on the features mentioned in the images. Still better quality is required to understand.
  10. In the conclusion, some literature gap is mentioned. Please address on the direction required to move further like, scope for future work. It is the mandatory outcome required for the quality journal.
  11. Please maintain uniformity in writing like, 175 m/s, 400m/s. In the first one, there is one-unit gap between numerals and unit, whereas in 2nd one no gap. Similarly, Reference nos. 21and 32, all letters of words in the title are upper case whereas it is different in others.
  12. Please do sentence check for grammatical mistakes.

Author Response

Dear Reviewer #3,
We wish to thank the Reviewer for acknowledging our work and recommending it for publication. We also wish to express our gratitude for all your comments on how to improve the manuscript. We have
taken all suggestions into account and made necessary revisions, as explained below. The revised fragments are marked in yellow. We hope that the manuscript is now more readable and coherent.
The main revisions made in the manuscript and our responses to the Reviewer’s comments are as
follows:
1. Reviewer #3: Please mention the following statement in the Abstract at the end as “Literature
gap leading to scope of future work is also included.”
Response: We have added the following sentence to the abstract.
2. Reviewer #3: Introduction, 2nd para seems to be like writing for text book, without any literature
support. As justifications, please add some literature support for the arguments.
Response: We have added more citations and revised the introduction.
3. Page 2, last sentence, “The effects of these treatments are in this work are described and the
results are presented after 3D printing of metallic components”. Please correct the grammatical
mistake.
Response: We apologise for the grammatical mistake. We improve this phrase as follows:
‘’Then, we presented results on SP, CP and the application of the aforementioned methods of
metallic material properties after additive manufacturing.’’
4. Page 3, 1st paragraph, “For this reason, various peening methods have been used for machining
parts made of metallic materials produced by printing and combinations of peening methods
with other machining methods, such as heat treatment, are also used.” There is ambiguity in
the sentence. How heat treatment is the machining method?
Response: We corrected this mistake as follow: “For this reason, various peening methods
have been used for finishing machining parts made of metallic materials produced by printing
and combinations of peening meth-ods with other methods, such as heat treatment, are also
used [16].”
5. I could not understand about how the figure 1 (Unary phase diagram of water) is fitted to the
present discussion? The phase transformation process is related to Gibb’s phase rule, where
the number of phases at equilibrium is connected to the variables and degrees of freedom.
Please explain. Otherwise remove figure 2.
Response: Figure 2. has been deleted.
6. A table, containing the types of cavitation, effect, benefits, type of surface damage quoting
published papers may be useful to the users. This can be added in the manuscript.
Response: As recommended, we have added a table describing selected aspects related to
cavitation peening (Table 3.).
7. Similarly, under shot peening, a table, containing the development of surface roughness,
hardness, corrosion resistance, fatigue resistance and fretting as per different works in the
domain by different researchers can be included to add value to the manuscript.
Response: As per comments, we have also included a table describing the findings of other
researchers in the field of shot peening (Table 2.).
8. Similarly, shot peening effects may be presented in the form of table, comprising of materials,
process parameters, property improved etc. to minimise monotonous approach of writing as
well as more data may be show cased.
Response: According to comments, we have also added a table describing the findings of other
shot peening researchers, and expanded the 6th paragraph to include current publications.
9. Figure 10, no clarity on the features mentioned in the images. Still better quality is required to
understand.
Response: As far as we could, we made our own arrangements for the figure 10.
10. In the conclusion, some literature gap is mentioned. Please address on the direction required
to move further like, scope for future work. It is the mandatory outcome required for the quality
journal.
Response: To better indicate future directions, we have given the following section in chapter
7 “The combination of CP and SP methods was suggested in connection with the search for
new solutions in the field of shot peening. Because different materials respond differently to shot
peening, especially after heat treatment, thermochemical treatment, and the use of different
manufacturing or processing methods. It is worth considering a combination of SP and CP, as
it can have positive effects in terms of the depth of the layer strengthened and the quality of the
surface and give greater control over the strengthening effect. A combination of these methods
can also be useful due to the surface cleaning effect of cavitation, which can have positive
effects on the reduction of cytotoxicity in biomaterials, and reduce the corrosion potential.”
and add in summary:
“The literature gap leading to the scope of future work seems a need to investigate the effects
of peening, especially cavitation peening and hybrid peening, on anti-wear and corrosion
performance of additively manufactured metallic materials.”
11. Please maintain uniformity in writing like, 175 m/s, 400m/s. In the first one, there is one-unit gap
between numerals and unit, whereas in 2nd one no gap. Similarly, Reference nos. 21and 32, all
letters of words in the title are upper case whereas it is different in others.
Response: Thank you for that remark. All these typos were improved and unified in the
manuscript.
12. Please do sentence check for grammatical mistakes.
Response: We have done the grammar check and eliminated the typos. We hope the
readability has been toughly improved.
Thank you for considering our manuscript for publication.
Yours sincerely,
Aleksander Świetlicki and Mirosław Szala
(corresponding authors)

Reviewer 4 Report

The paper : Effects of shot peening and cavitation peening on properties of surface layer of metallic materials - a short review present some data from an interesting field that might have many industrial applications in the near future.

However the authors must work on the material in order to be publish because: the abstract must be carefully re-written in order to transmit the message of the review article and what the authors follow to highlight;  the introduction is very poorly done : there are no references in the introduction section; where are the images and graphs from Figure 1 comment in text ? appreciations must be done on the left side figures from figure 1 and what does represent, the scale are not visible and why are they contradictory? left images with the graphs from the right ?. This section must be entirely re-written. 

Some minor concerns:

at table 1 explanation loose the third comma, 

re-phrase Figure 2 explanation 

mention Hydraulic cavitation as 2.1 and Acoustic cavitation as 2.2 

Figure 8 has a low quality 

in Summary section give some perspectives of the subject 

explain PBF acronym 

103 .. - 103.

the references are on the subject, covering a large period of time 

all figures/images need an improve quality version 

the conclusions must be restructured 

Author Response

Dear Reviewer #4,
We wish to thank the Reviewer for acknowledging our work and recommending it for publication. We also wish to express our gratitude for all comments on how to improve the manuscript. We have taken all suggestions into account and made necessary revisions, as explained below. The revised fragments
are marked in yellow. We hope that the manuscript is now more readable and coherent.
The main revisions made in the manuscript and our responses to the Reviewer’s comments are as
follows:
Reviewer #4: The paper : Effects of shot peening and cavitation peening on properties of surface
layer of metallic materials - a short review present some data from an interesting field that might have
many industrial applications in the near future.
Reviewer #4: However the authors must work on the material in order to be publish because:
the abstract must be carefully re-written in order to transmit the message of the review article
and what the authors follow to highlight;
Response: Thank you for that remark. The abstract was rewritten entirely.
Reviewer #4: the introduction is very poorly done: there are no references in the introduction
section;
Response: We have added more citations and revised the idea of the introduction section.
Reviewer #4: where are the images and graphs from Figure 1 comment in text ? appreciations
must be done on the left side figures from figure 1 and what does represent, the scale are not
visible and why are they contradictory? left images with the graphs from the right ?. This section
must be entirely re-written.
Response: The fig 1 has been deleted. The introduction section has been completely revised.
Reviewer #4: Some minor concerns:
at table 1 explanation loose the third comma,
Response: We improved the caption of table 1.
Reviewer #4: re-phrase Figure 2 explanation
Response: According to the recommendation of the other reviewer, figure 2 was deleted.
Reviewer #4: mention Hydraulic cavitation as 2.1 and Acoustic cavitation as 2.2
Response: We improved the arrangement of the sections. It was revised as follows:
4.1.1 Hydrodynamic cavitation
4.1.2 Acoustic cavitation
Reviewer #4: Figure 8 has a low quality
Response: The quality of figures 3, 7(8) and 9 has been improved.
Reviewer #4: in Summary section give some perspectives of the subject
Response: The summary section has been improved according to your recommendation.
Reviewer #4: explain PBF acronym
Response: Thank you for that remark. The PBF acronym has been explained in the text as
follows:
Figure 8. shows the mechanism of particle removal from the surface of produced by means of
powder bed fusion (PBF) under cavitation conditions.
Reviewer #4: 103 .. - 103. the references are on the subject, covering a large period of time
Response: Thank you for that remark. According to the rev#2 we updated some of the
references to make the paper much more current.
Reviewer #4: all figures/images need an improve quality version
Response: The quality of figures 3, 7 and 9 has been improved.
Reviewer #4: the conclusions must be restructured
Response: We have restructured the final section of the paper.
Thank you for considering our manuscript for publication.
Yours sincerely,
Aleksander Świetlicki and Mirosław Szala
(corresponding authors)

Round 2

Reviewer 4 Report

Publish in the current form.